# Sestrin3 confers resistance to recombinant human arginase in small cell lung cancer by activating Akt/mTOR/ASS1 axis

Zhongqiang Zhang[1,2‡], Zizhe Lin[1,2‡], Weishan Li[1], Binxiong Chen[1], Yueming Liu[1], Hanchao Gao[2,3]*, Shi Xu[1,2]*

1 Department of Burn and Plastic Surgery, Shenzhen Longhua District Central Hospital, Shenzhen, Guangdong, China, 2 Department of Medical Laboratory, Shenzhen Longhua District Central Hospital, Shenzhen, Guangdong, China, 3 Department of Nephrology, Shenzhen Longhua District Central Hospital, Shenzhen Longhua District Key Laboratory for Diagnosis and Treatment of Chronic Kidney Disease, Shenzhen, Guangdon, China

‡ These authors contributed equally to this work and share first authorship.
* xushi_cn@163.com (SX); hcgao@foxmail.com (HG)

## Abstract

Drug resistance is a major obstacle in the clinical management of small cell lung cancer (SCLC), we have proved the promising anticancer effect of recombinant human arginase (rhArg, BCT-100) in SCLC *in vitro* and *in vivo*. In order to promote the clinical application of recombinant human arginase, it is necessary to explore the underlying resistant mechanisms of BCT-100 in SCLC. Here, we cultured and obtained the acquired drug-resistant SCLC cell line (H446-BR), which displayed different cellular phenotypes (enhanced migration ability) compared with the parental cell line (H446). sestrin3 (SESN3) was confirmed with high expression in resistant cell line. Knockdown SESN3 could re-sensitize resistant cells to BCT-100 treatment and reverse the aggressive feature of H446-BR. The Akt-mTOR signal pathway and ASS1, which were highly expressed in resistant cells, were down-regulated after silencing SESN3. MK-2206 and rapamycin suppressed the expression of ASS1 in H446-BR cell. In xenograft model, BCT-100 has little anti-tumor effect on H446-BR compared with H446 as well as H446-BR silenced sestrin3. Collectively, these results elucidate SESN3 plays an essential role in resistant mechanism, which will provide a valuable source of information for translational research.

## Introduction

Lung cancer ranks the top of malignant tumor incidence rate for a long time and poses a serious threat to people's health. Although small-cell lung cancer (SCLC) only accounts for 15% of the total number of lung cancers, this malignancy is prone to relapse and easy to develop drug resistance. The five-year survival rate of SCLC patients with relapse is less than 8%, which is considered as one of most deadly

**Data availability statement:** All relevant data are within the paper and its Supporting Information files.

**Funding:** This research was supported by the Scientific Research Projects of Medical and Health Institutions of Longhua District, Shenzhen (Grant No. 2022015 & 2022131& 2024024) awarded to ZZ and ZL, Shenzhen Longhua District Science and Technology Innovation Special Fund Project (Grant No. 11501A20220923BF59236 & 11501A20220923BE5B6B3 & 11501A20220923BD5F291& 11501A20240704B338532) awarded to HG and SX, Medical Scientific Research Foundation of Guangdong Province (Grant No. A2025230), and Guangdong Basic and Applied Basic Research Foundation (Grant No. 2024A1515012821) awarded to SX. All these funders had no role in study design, data collection and analysis, decision to publish, or preparation of the manuscript.

**Competing interests:** NO authors have competing interests.

cancers with high malignancy and mortality [1–3]. Previously, recombinant human arginase (BCT-100) has been demonstrated as a promising anticancer drug on SCLC. To facilitate future clinical application of BCT-100, we explored the underlying resistant mechanism of BCT-100 on SCLC.

As a semi-essential amino acid, arginine could be independently synthesized by human cell under normal physiological conditions. However, some arginine-auxotrophic cancer cells cannot autonomously synthesize arginine due to lacking of argininosuccinate synthetase 1 (ASS1) or ornithine transcarbamylase (OTC). These cells have to rely on the extracellular arginine to grow. Therefore, arginine depletion agent has been considered as a targeted anti-tumor drug due to its relative inert to normal human cells, but great anti-tumor activity on arginine-auxotrophic cancer cells [4]. Currently, there are two arginine depletion agents used in clinical or pre-clinical studies, one is arginine deiminase (ADI) and the other is recombinant human arginase (rhArg) [5,6]. ADI, obtained from mycoplasma, can directly catalyze arginine to its precursor citrulline, leading intracellular arginine depletion [7]. While, in line with endogenous arginase, rhArg transforms arginine to ornithine and urea, thus reducing arginine level. BCT-100 was pegylated with polyethylene glycol, which could enhance its catalytic activity, increase the half-life, and reduce immunogenicity [8]. Mounting evidences have revealed that the promising anti-cancer effects of rhArg on various cancer cells including liver cancer, leukemia, SCLC, and pancreatic cancer [5,9–11]. However, drug resistance still impedes further application of rhArg in clinical practice, especially for SCLC with susceptibility to drug resistance.

Sestrin3 (SESN3) belongs to a highly conserved family of stress-induced proteins, mainly consisting of three subfamilies (SESN1, SESN2 and SESN3), which have a variety of cell biological functions including regulating DNA damage, oxidative stress, hypoxia and aging [12]. SESN1 is also known as P53 regulatory protein PA26, and SESN2 is a PA26 homologue. SESN3 is a newly discovered structure-related protein of PA26 [13]. It was found that SESN1 and SESN2 could regulate the state of cells to oxidative stress, ER stress and hypoxia by activating AMPK and inhibiting mTOR signaling pathway, and the growth of lung cancer cells in nude mice can be inhibited by inactivating Akt-mTOR cell signaling pathway after SESN2 deletion. Previous studies have confirmed that SESN2 regulates energy metabolism via regulating the AMPK/mTOR signaling pathway, as well as survival and apoptosis in sensory hair cells under stress [14,15]. Besides, SESN2 can simultaneously activate AKT and AMPK signaling pathway to mediate the drug resistance of liver cancer cells to sorafenib [16]. The biological functions of SESN3 were preliminarily explored. SESN3 plays an important role in the anti-tumor process of cucurbitinoid B, and silencing SESN3 can reduce the proliferation of tumor cells and cause cell cycle G2 arrest, and promote the apoptosis induced by cucurbitinoid B [17]. It has been demonstrated that SESN3 could regulate pre-adipocytes adipogenesis via inhibiting SMAD3 [18]. These evidences indicate that SESN family has a broad application prospect and value in anti-tumor and drug resistance research, but research on SESN3 is still in backward stage, and further in-depth research is needed.

In present study, our team members aim to explore the underlying mechanisms of acquired resistance to rhArg in SCLC. Here, we found SESN3 was highly expressed in resistant cells, and Akt/mTOR/ASS1 axis accounts for the acquired resistance to rhArg in SCLC. These findings might provide novel insights into drug resistance of rhArg and scientific ground for future clinical development in SCLC.

## Materials and methods

### Cell lines and cell culture

Small cell lung cancer cell line NCI-H446 was purchased from the American Type Culture Collection (Manassas, VA, USA). And the corresponding BCT-100 resistant cell line H446-BR was established by exposing with stepwise increasing concentrations of BCT-100 for a long term as described in previous study [19]. Both cell lines were maintained in RPMI-1640 medium (Gibco, Life Technologies, Carlsbad, California, USA) supplemented with 10% fetal bovine serum (FBS) (Meilunbio, Dalian, China) in a humidified atmosphere containing 5% $CO_2$ at 37°C.

### Reagents

BCT-100 was kindly provided by Bio-cancer Treatment International Limited, P.R.China, HKSAR. Cisplatin (Cat# 232120) and etoposide (Cat# 341205) were both purchased from Sigma Aldrich. MK-2206 (Cat# SF2712−5 mg), as well as rapamycin (Cat# S1842-25 mg), was bought from Beyotime Biotechnology (Jiangsu, China).

### Western blot analysis

Western blot assay was conducted as previously described [20]. Primary antibodies used in this study were displayed in Table 1. Quantification was conducted using Image J software (National Institutes of Health, Bethesda, MD, USA).

### RNA isolation and real-time polymerase chain reaction (RT-qPCR) assay

TRIzol™ Reagent (Invitrogen, Cat# 15596026CN) was employed for RNA extraction. The High Capacity cDNA Reverse Transcription kit (Applied Biosystems, 4374966) and PowerUp™ SYBR™ Green Master Mix (Applied Biosystems, A25741) were used to perform Reverse transcription of RNA and RT-qPCR, respectively. The steps were based on the manufacturer's protocol. The relative gene expression in the samples was calculated by $2-\Delta\Delta CT$ method as previously delineated in previous literature [21]. The forward and reverse primers for humans tested were as follows:

**Table 1. List of primary antibodies used in this study.**

| Name | Species | Manufacture | Cat# | Molecular weight (kDa) | Dilution factor |
|------|---------|-------------|------|------------------------|-----------------|
| ASS1 | Rabbit | Santa Cruz Biotechnology | SC-99178 | 47 | 1:500 |
| C-PARP | Rabbit | Santa Cruz Biotechnology | SC-7150 | 89 | 1:1000 |
| β-Actin | Mouse | Beyotime Biotechnology | AA128 | 42 | 1:1000 |
| SESN3 | Mouse | Santa Cruz Biotechnology | SC-517092 | 57 | 1:200 |
| m-TOR | Rabbit | Cell Signaling Technology | 2972 | 289 | 1:1000 |
| p-mTOR | Rabbit | Cell Signaling Technology | 5526 | 289 | 1:1000 |
| C-Caspase3 | Rabbit | Cell Signaling Technology | 9664 | 17 | 1:1000 |
| ALDH1 | Rabbit | Cell Signaling Technology | 36671 | 55 | 1:1000 |
| Nanog | Rabbit | Cell Signaling Technology | 8822 | 42 | 1:1000 |
| Oct-4 | Rabbit | Cell Signaling Technology | 2750 | 45 | 1:1000 |
| AKT | Rabbit | Cell Signaling Technology | 9272 | 60 | 1:1000 |
| p-AKT | Rabbit | Cell Signaling Technology | 4060 | 60 | 1:1000 |

SESN3-F: 5'- CTCCCAGACAGGCACTACATT −3',
SESN3-R: 5'- TAGGTAGGAACACTGGTGTCTGG −3',
GAPDH-F: 5'-GGTGGTCTCCTCTGACTTCAACA-3',
GAPDH-R: 5'-AGCTTGACAAAGTGGTCGTTGAG-3'.

## Cell viability assay

Cells were cultured in a 96-microplate at a density of approximately $10^4$ cells per well. After drug exposure, cells were incubated with 10 μl of CCK-8 solution (Abcam, Cat# ab228554) for 1–4 hours at an incubator protecting from the light. Optical density was measured at 450 nm using Fluo Star Optima microplate reader (BmgLabtec GmbH, Ortenberg, Germany).

## Short hairpin RNA (shRNA) transfection

Silencing of SESN3 (shRNA) was performed by employing the lentiviral particles kit purchased from Santa Cruz (Cat# sc-106545-V), and the sequences were not provided due to business confidential work. The step was carried out according to the manufacturer's protocol and was previously delineated in previous work [20]. The protein and mRNA expression of SESN3 was evaluated by Western blot and qRT-PCR assay, respectively.

## Flow cytometry of apoptosis detection

Cellular apoptosis was measured by flow cytometry using Annexin V-FITC apoptosis detection kit (Beyotime Biotechnology, Cat# C1062S). Briefly, cells were harvested, washed, and resuspended in binding buffer at $5 \times 10^6$ ml$^{-1}$. FITC-conjugated Annexin V solution (5 μl) and PE-conjugated PI solution (10 μl) were added to 100 μl of cell suspension. After incubation with annexin V/PI for 15 min at room temperature, samples were measured using BD FACSAriaII analyzer with FL2/FL4 channels (BD, New Jersey, USA). Every sample counted 10000 events for quantification.

## Flow cytometry of reactive oxygen species (ROS) measurement

The presence of hydrogen peroxide ($H_2O_2$) was determined by using ROS Assay Kit bought from Beyotime Biotechnology (Cat# S0033M). Generally, treated cells were incubated with H2DCFDA (1 μM) in medium without FBS for 30 min at 37˚C, followed by washing twice before being analyzed by flow cytometry (BD FACSAriaII analyzer) to quantify ROS production.

## Flow cytometry of mitochondrial membrane depolarization (MMD)

Mitochondrial membrane potential (ΔΨm) was measured using enhanced mitochondrial membrane potential assay kit with JC-1 (Beyotime Biotechnology, Cat# C2003S). In brief, specific samples were harvested, washed with PBS, followed with incubating in FBS-free medium with JC-1 (5 μg/ml) at 37°C for 20 min in darkness. PBS was used to wash treated cells to decrease background fluorescence. Relative JC-1 fluorescent signal intensity was tested by flow cytometry in FL1/FL2 channels.

## Tumorigenicity evaluation

Xenograft model was used to evaluate the tumorigenicity of H446-BR cell. All animal assays were performed in compliance with guidelines approved by the Animal Use and Care Administrative Advisory Committee at Shenzhen Longhua District Central Hospital and the laboratory animal ethical committee of Guangdong Medical University (GDY2202302). The procedure was previously delineated [20]. Xenograft model was established by s.c. injection of $1*10^6$ (100 μl) cells into the upper dorsum of nude mice. The frequency of tumor monitoring once palpable is twice weekly. Tumor size was calculated according to the formula: (Size = Length × Width × Depth/2) [22]. Mice were euthanized using sodium pentobarbital (i.p.,

180 mg/kg) when the tumor volume over 800 mm³, which was regarded as a humane endpoint. Once animals reached endpoint criteria, the amount of time elapsed before euthanasia was less than 10 min. Cervical dislocation was used to confirm animal death without breathing or heartbeat. Animal health and behavior were monitored daily by trained staff to ensure well-being. No unexpected deaths occurred during the study. The number of mice used in every group was 8, and the duration of the experiment last 40 days.

### EdU labeling assay

Cell proliferation assay was conducted using BeyoClick™ EdU Cell Proliferation Kit with Alexa Fluor 488 (Beyotime, Cat# C0071S) according to the standard protocol provided by the manufacturer. The data were collected using Zeiss fluorescence microscope.

### Statistical analysis

All data were obtained from at least three independent experiments and are expressed as mean±standard deviation (SD). Student's two-tailed t-test was used for comparison of pairs. The difference between groups (more than two groups) was analyzed using variance analysis (ANOVA) with the post hoc test (Tukey's Test) or Mann-Whitney $U$ test by Prism 5 (GraphPad Software, La Jolla, Southern California, USA).

## Results

### H446-BR cells displayed resistance to BCT-100 and chemotherapy agents

After long term of cell culture with gradually increasing level of BCT-100, the parental cell line H446 acquired drug resistance to BCT-100 and termed as H446-BR, which showed strong resistance to BCT-100 (Fig 1A). Apart from resistance to BCT-100, H446-BR also displayed resistance to first-line chemotherapy, cisplatin and etoposide, compared with parental cells (Fig 1B and 1C).

### Stemness related biomarkers were induced in H446-BR cells with aggressive migration ability

To investigate the feature of resistant cells, wound healing assay was used to the migration ability. Compared with the parental cell line, H446-BR obtained more aggressive migration ability as showed in Fig 2A and 2B. Besides, the pluripotent stem cell biomarkers such as ALDH1, Oct-4, and Nanog, were highly expressed in H446-BR cells (Fig 2C and 2D).

### SESN3 silencing inhibited the cellular migration and promoted cellular apoptosis

SESN3 was markedly up-regulated in resistant cells both in mRNA level and protein level (Fig 3A and 3B). To further confirm the key role of SESN3 in resistant mechanism, H446-BR cells were infected with either negative control shRNA or specific shRNA to knockdown SESN3. We found that silencing SESN3 could attenuate the migration ability in H446-BR cells compared to control group (Fig 3C and 3D). Besides, EdU staining assay was used to determine the proliferation of resistant cells treated with BCT-100 after SESN3 silencing, which suppressed the proliferation of H446-BR as shown in Fig 3E. In line with previous results, ALDH1, Oct-4, Nanog, and N-cadherin were suppressed in H446-BR cell line infected with SESN3 (Fig 3F). According to the results of cell viability assay, we found that BCT-100 (20 mU/ml) could also induce the apoptotic biomarker cleaved poly-ADP-ribose polymerase (C-PAPR) as displayed in Fig 3G.

### Knockdown of SESN3 inhibits BCT-100-resistant cell proliferation in vivo

To evaluate whether SESN3 silencing could suppress SCLC tumor growth in vivo, H446-BR cells infected with shNEG, shSESN3−1 or shSESN3−2 were used to establish model of transplanted subcutaneously tumors in nude mice. The tumor size derived from silenced arms (shSESN3−1 or shSESN3−2) were significantly decreased compared to those

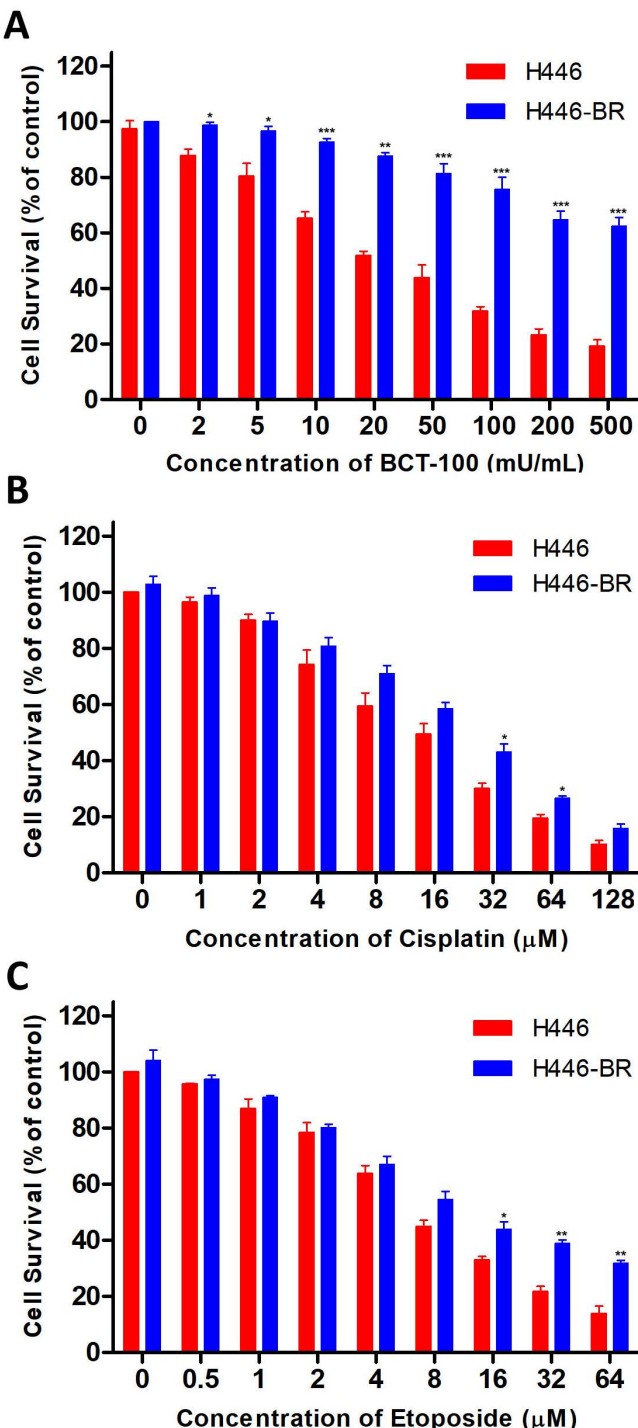

**Fig 1. Cell viability of H446-BR cells to BCT-100 and chemotherapy agents.** (A) Cell growth inhibition evaluated by CCK-8 assay in H446 and H446-BR cells treated with BCT-100 for 72h. (B) Cisplatin and etoposide (C) were treated to H446 and H446-BR cells for 72h, respectively. Error bars indicate the mean (SD); n = 3. P-values were determined using one-way ANOVA (*$P < 0.05$, **$P < 0.01$, ***$P < 0.001$).

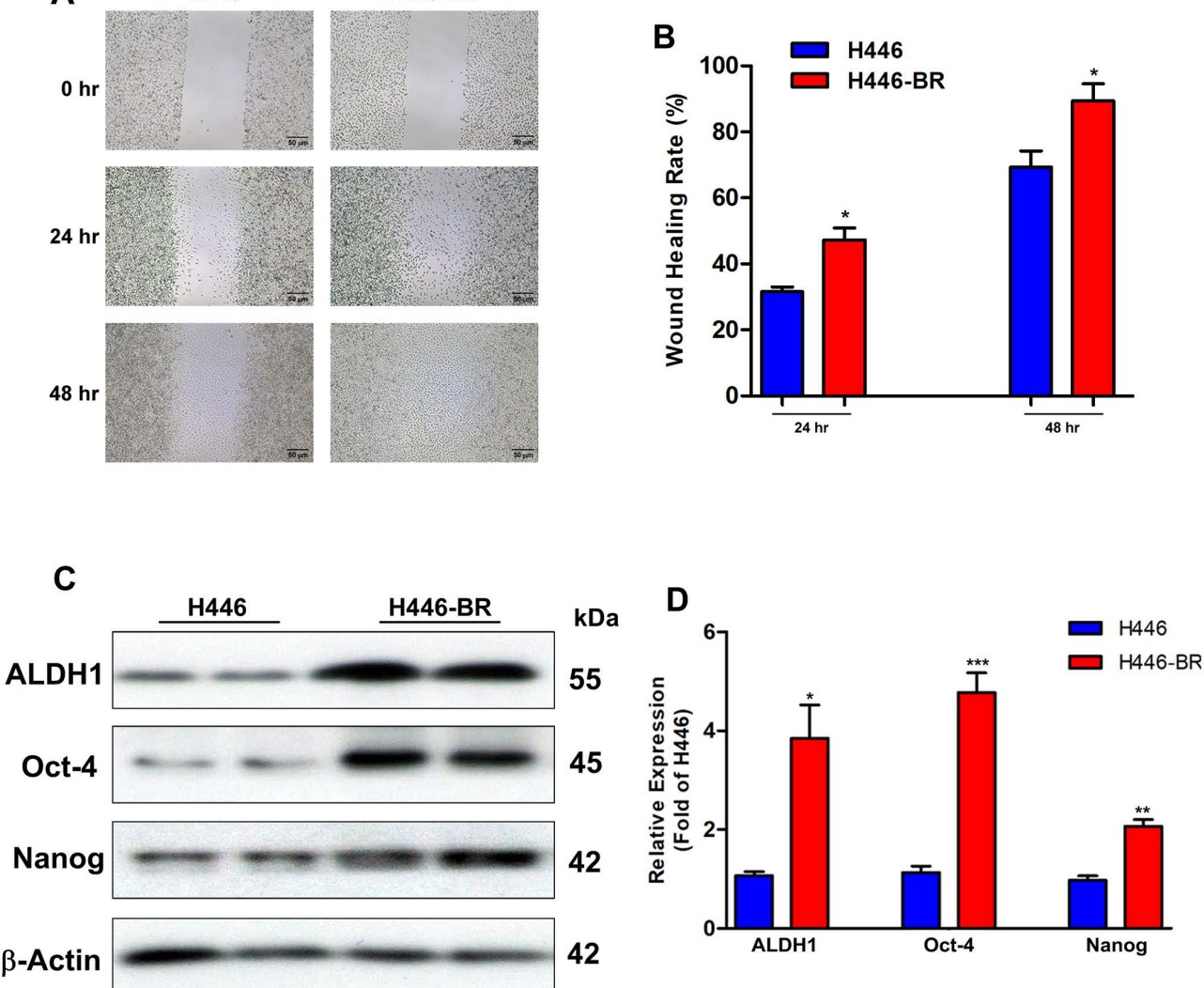

**Fig 2. H446-BR cells displayed fast-growing phenotype with strong migration ability.** (A) Wound healing assay to assess cellular migration ability of H446 and H446-BR. (B) Quantitation of migration rate (24 hr and 48 hr) in H446 and H446-BR cells. (C) Western blot to evaluate the basal expression of ALDH1, Oct-4, and Nanog in H446 and H446-BR cells. (D) Quantitation of ALDH1, Oct-4, and Nanog in H446 and H446-BR cells. β-Actin was used as the loading control, and error bars indicate the mean (SD); n = 4. P-values were determined using one-way ANOVA (*$P < 0.05$, **$P < 0.01$, ***$P < 0.001$).

in control group (Fig 4A and 4B). Notably, BCT-100 (60 mg/kg) could enhance the anti-tumor effect and reduce tumor volume greatly in silenced arm (Fig 4A). Consequently, the median survival of mice was prolonged from 25.0 days (control group) to 32.0 days (shSESN3−1 arm) as shown in Fig 4C. In line with previous experiment, the protein expression of SESN3 was significantly reduced in treatment arm (Fig 4D). In our previous study, contactin-1 was identified as a key gene for resistance. To confirm the synergistic effect of SESN3 and contactin-1 in sensitizing the resistant cells, we performed a double knockout of these two genes. Notably, the wound healing rate in double knockout group (35.0 ± 13.2%) was lower than single knockout group (66.3 ± 3.5% and 70.7 ± 12.4%) as shown in S1 Fig.

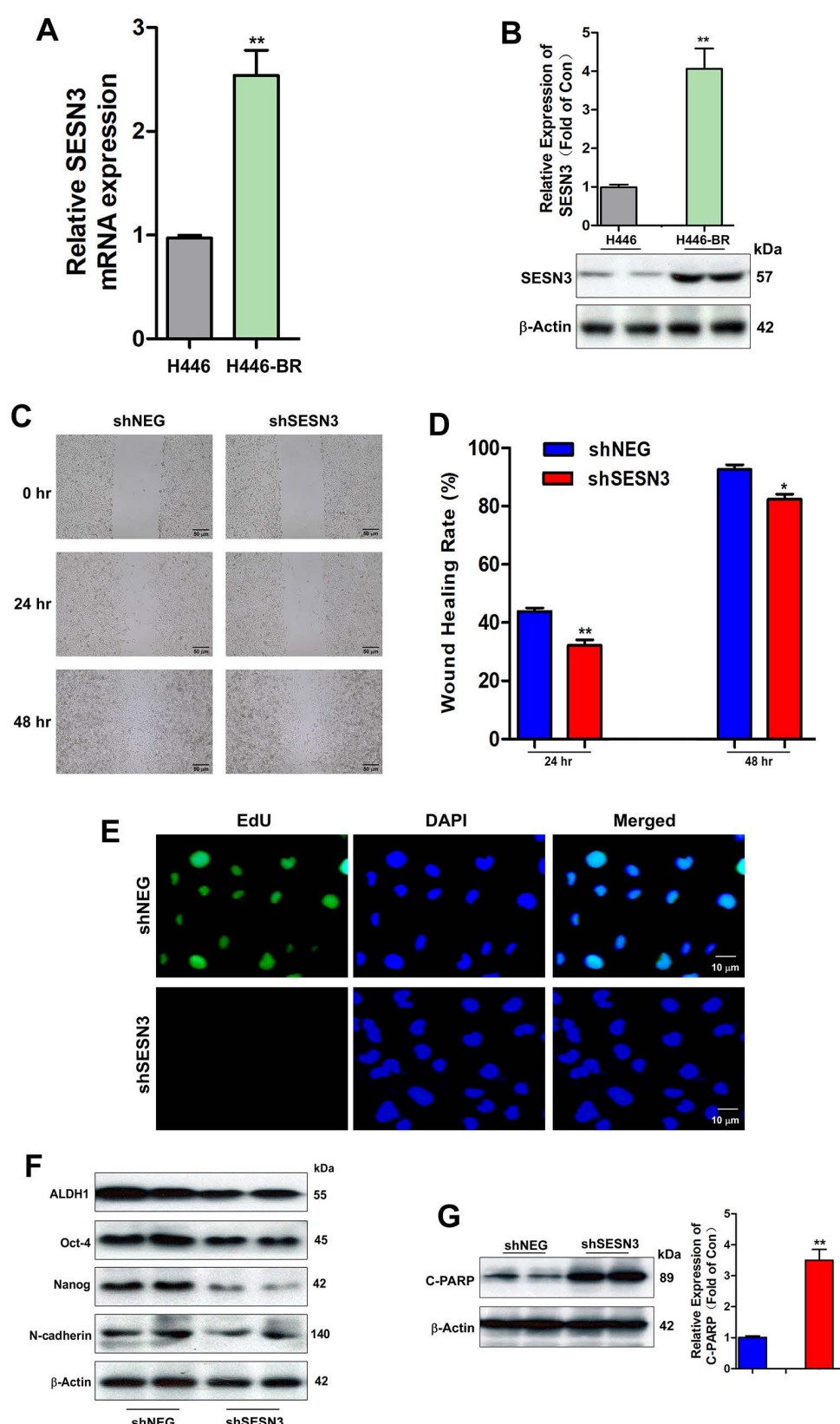

**Fig 3. The high expression of SESN3 in resistant cells accounts for strong migration and proliferation.** (A & B) Elevation of mRNA and protein level of SESN3 H446 and H446-BR cells analyzed by RT-PCR and Western blot, respectively. (C) Wound healing assay to assess cellular migration ability of H446-BR infected with shNEG or shSESN3 at different time points (0 hr, 24 hr, 48 hr). (D) Quantitation of wound healing rate H446-BR infected with shNEG or shSESN3 at 24 hr and 48 hr. (E) EdU staining to test the cell proliferation of H446-BR infected with shNEG or shSESN3 exposing BCT-100 (20 mU/ml) for 3 days. (F) Western blot to evaluate ALDH1, Oct-4, Nanog, and N-cadherin in H446-BR cells infected with shNEG or shSESN3. (G) Western blot to evaluate the apoptotic biomarker (C-PARP) in H446-BR cells infected with shNEG or shSESN3 exposing BCT-100 (20 mU/ml) for 3 days. β-Actin was used as the loading control. Error bars indicate the mean (SD); n = 4. P-values were determined using one-way ANOVA (*$P < 0.05$, **$P < 0.01$, ***$P < 0.001$).

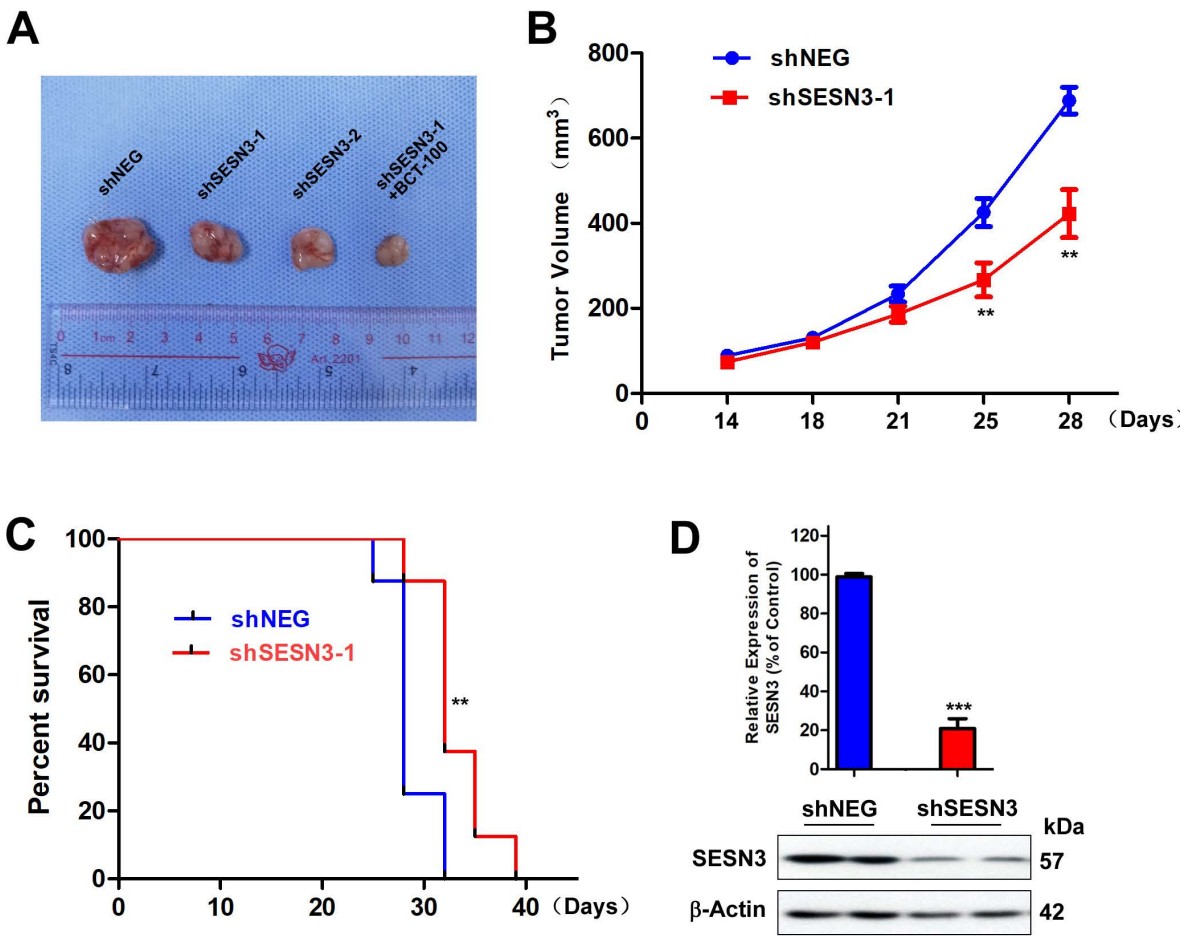

**Fig 4. SESN3 knockdown suppresses tumor growth in vivo.** (A) Representative photos of excised tumors after treatment with shNEG, shSESN3−1, shSESN3−2 and BCT-100 + shSESN3−1 at 20 days post-treatment. (B) Tumor volume and (C) median survival of mice in the shNEG and shSESN3−1 groups recorded in different time points. Error bars indicate the mean (SEM); n = 8. P-values were determined using a Mann-Whitney *U* test. (D) The intratumoral protein level of SESN3 in xenograft model was tested by Western blot. β-Actin was used as the loading control. Error bars indicate the mean (SD); n = 8. P-values were determined using an unpaired Student's t-test (***$P < 0.001$).

## SESN3 silencing promotes apoptosis through ROS overproduction and mitochondrial dysfunction in BCT-100-resistant cells

Our results above proved that SESN3 played an essential role in resistant mechanism. To further confirm the effect of SESN3, we analyzed cell apoptosis using Annexin V/PI double staining by flow cytometry. In the presence of BCT-100

(20 mU/ml), the relative apoptotic rate of shSESN3–1 (25.5±5.2%) or shSESN3–2 (24.7±5.6%) was much higher than that of control group (shNEG, 5.2±0.95%) (Fig 5A and 5B). Mitochondrial integrity was determined by flow cytometry with JC-1 staining, which was widely used to indicate the mitochondrial membrane potential. As displayed in Fig 5C and 5D, the relative green fluorescence intensity, which suggests JC-1 monomer ratio, was significantly higher in shSESN3–1 or shSESN3–2 group (20.6±9.0% or 18.9±3.6%) compared to shNEG group (3.5±1.9%). Meanwhile, in presence of 20 mU/ml BCT-100, ROS production of SESN3 silencing arms (25.9±4.7% or 24.2±6.9%) was higher than that of shNEG group (5.9±1.2%) as displayed in Fig 5E and 5F. To clarify the causal relationship between ROS and mitochondrial dysfunction, we used NAC, the ROS scavenger, to protect the resistant cells with SESN3 silenced. As displayed in S2 Fig, NAC could significantly reduce the JC-1 monomer ratio, suggesting less mitochondrial dysfunction. Consistent with previous study, the apoptotic biomarkers including C-PARP and C-Caspase3 were up-regulated in shSESN3–1 or shSESN3–2 arms compared to control group treating with BCT-100 for 3 days (Fig 5G and 5H).

### SESN3 regulates BCT-100-resistance via activating Akt/mTOR/ASS1 axis

Since Akt-mTOR signaling pathway plays crucial functions in cell growth, proliferation, and metabolism, and ASS1 was recognized as the key role in arginine metabolism, we investigated the Akt-mTOR signaling pathway and ASS1 in H446-BR cells. As displayed in Fig 6A and 6B, the basal expressions of p-Akt and p-mTOR in H446-BR cells were significantly higher than those of in parental cells. Notably, knockdown SESN3 in H446-BR could inhibit the expressions of p-Akt, p-mTOR, and ASS1 (Fig 6C and 6D). To further confirm the function of Akt/mTOR/ASS1 axis, both Akt inhibitor MK-2206 and mTOR inhibitor rapamycin exert inhibitory effect on ASS1 expression (Fig 6E and 6F). Supplementary experiment was performed to confirm the role of SESN3. SESN3 was overexpressed in H446 cells, and ASS1 was slightly induced (S3 Fig). SESN3 overexpression displayed resistant phenotype to BCT-100 treatment (S3 Fig). Besides, A mechanistic rescue assay was established by Akt activator SC79 in H446-BR cells infected with shSESN3, with recovery of p-Akt, p-mTOR, and ASS1 (S4 Fig).

## Discussion

Recombinant human arginase has recently emerged as a potentially promising anti-tumor agent in arginine auxotrophic cancer cells. Although the underlying anti-cancer mechanisms of rhArg in these cancer cells have been gradually demonstrated, the drug resistance still blocks the further application of rhArg. Thus, it is necessary to explore the drug resistance especially in SCLC. In our study, we found that high expression of SESN3 in H446-BR cells played an important role in drug resistant mechanism, which was involved in hyperactivity of Akt/mTOR/ASS1 axis.

Arginine depletion agents have significant efficacy in the treatment of arginine dystrophy type of cancer. Similar with chemotherapy drugs, drug resistance also restricts their further clinical application in the long term. The confirmed mechanisms of ADI resistance include reactivated expression of ASS1, enhancement of glycolysis and production of corresponding antibodies in vivo [23,24]. The basal expression of ASS1 is particularly low in malignant melanoma, however the half inhibitory concentration ($IC_{50}$) and ASS1 are significantly increased after ADI treatment for a long time [24]. In line with previous study, we also found that the expression of ASS1 in drug-resistant cells is significantly increased, which implied the key role of ASS1 in arginine metabolism and arginase resistance. Therefore, ASS1 can be used as a good marker of treatment effect and drug resistance of arginine depleting agents, which is worthy of further study. In addition, Long et al pointed out that ADI drug-resistant cell lines of melanoma, the expression of lactate dehydrogenase A and glucose transporter 1 was increased, while the expression of pyruvate dehydrogenase was decreased, indicating the glycolytic pathway was activated [24]. Although ADI is modified with polyethylene glycol to reduce its immunogenicity, ADI's antibodies are still detected in clinical trials [25]. Compared with ADI resistance, the study of rhArg resistance is still limited. Our previous study found that contactin 1, which is an adhesion molecule involved in carcinogenesis and cancer progression, accounts for drug resistance in SCLC by inducing Akt signaling pathway accompanied by epithelial–mesenchymal transition progression [19].

 

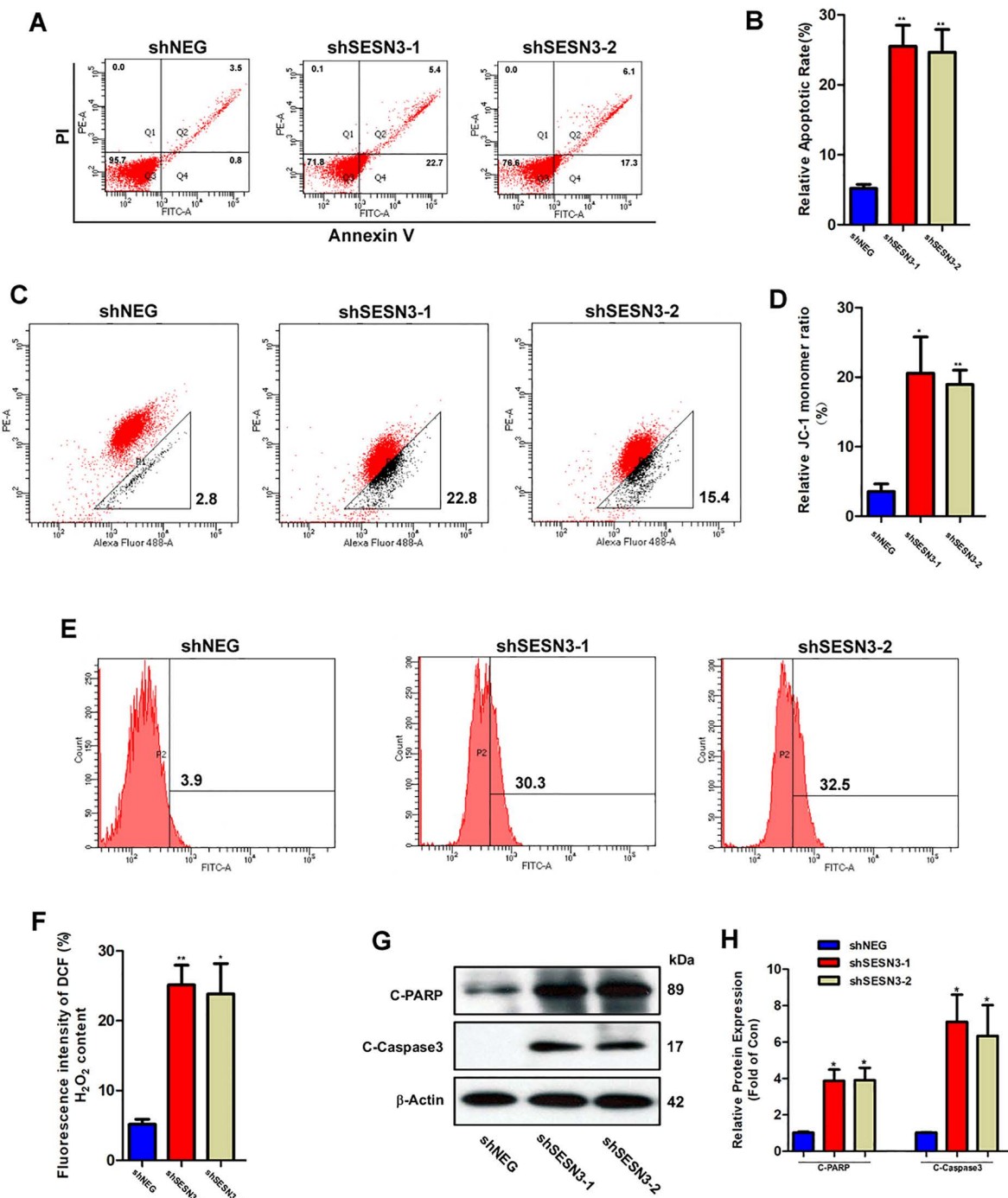

**Fig 5. SESN3 silencing induces apoptosis via ROS production and mitochondrial dysfunction in resistant cells.** (A) Representative histograms of cell apoptotic rate determined by flow cytometry of Annexin V/PI staining in SESN3 silenced H446-BR cells treated with BCT-100 (20 mU/ml) for 3 days. (B) Quantitation of cell apoptotic rates in shNEG, shSESN3-1, and shSESN3-2 group. (C) JC-1 staining measured by flow cytometry in SESN3 silenced H446-BR cells treated with BCT-100 (20 mU/ml) for 3 days. (D) Quantitation of JC-1 monomer ratio in corresponding groups. (E) ROS levels determined by flow cytometry in H446-BR cells infected with shNEG, shSESN3-1, or shSESN3-2, following with incubation with BCT-100 (20 mU/ml, 3 days). (F) Quantitation of ROS levels in corresponding groups. (G) Protein expression of C-PARP, C-Caspase3, and β-Actin evaluated by Western blot. (H) Quantitation of relative levels of C-PARP and C-Caspase3 analyzed by Image J software. β-Actin was used as loading control. Error bars indicate the mean (SD); n = 3. P-values were determined using one-way ANOVA (*$P < 0.05$, **$P < 0.01$, ***$P < 0.001$).

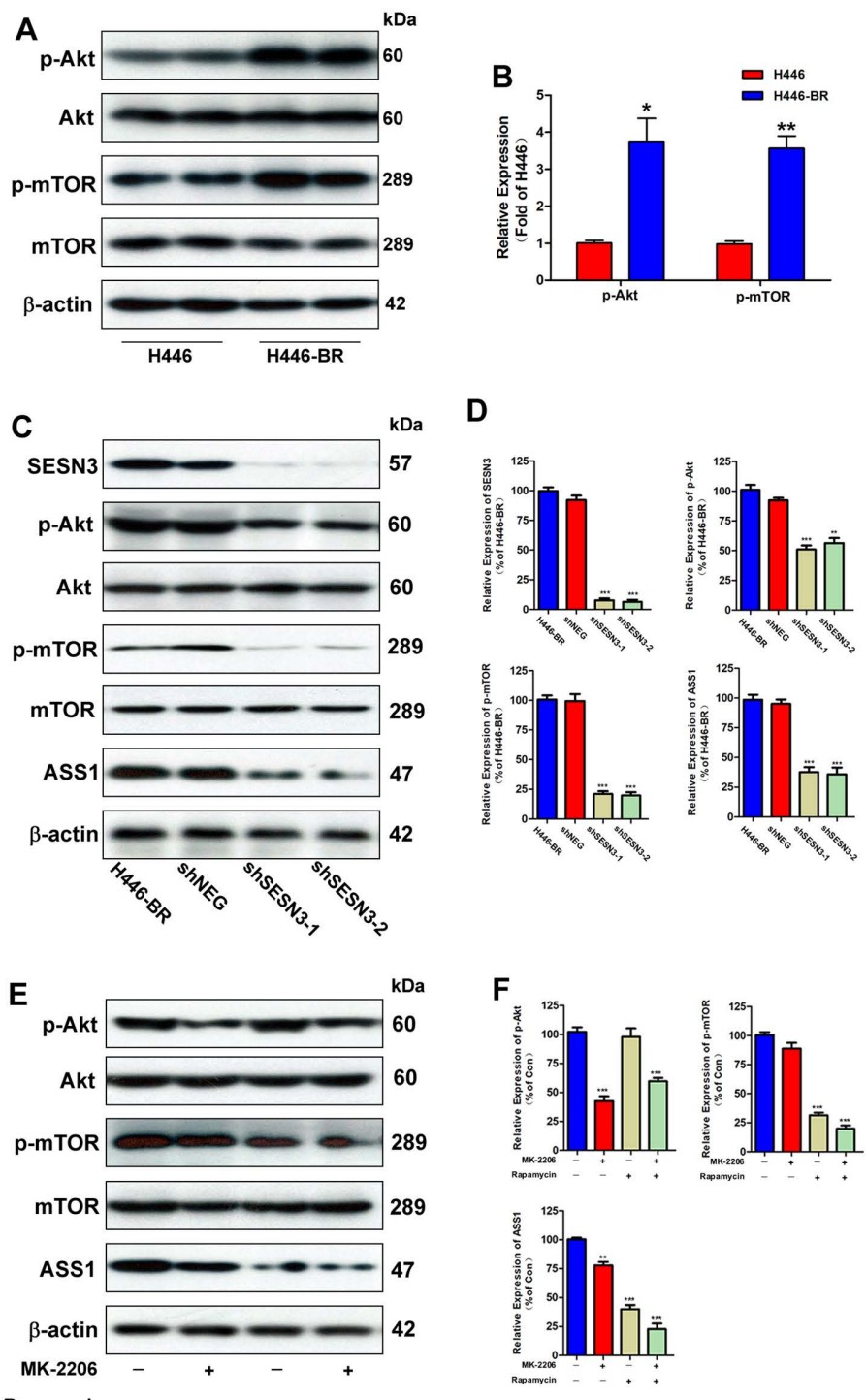

**Fig 6. Sestrin3 confers drug resistance through inducing Akt/mTOR/ASS1 axis.** (A) The basal expressions of Akt, p-Akt, mTOR, p-mTOR, and β-Actin were assessed by Western blot in H446 and H446-BR cells. (B) Quantitation of relative levels of p-Akt, and p-mTOR analyzed by Image J software. (C) Expressions of SESN3, p-Akt, p-mTOR, ASS1, and β-Actin were assessed by Western blot in H446-BR, H446-BR infected with negative control shRNA, shSESN3-1, and shSESN3-2. (D) Quantitation of SESN3, p-Akt, p-mTOR, and ASS1 in the corresponding groups. (E) Levels of related proteins tested by Western blot in H446-BR cells treated with MK-2206 (2 μM) or rapamycin (100 nM) with BCT-100 (20 mU/ml) for 3 days. (F) Quantitation of p-Akt, p-mTOR, and ASS1 analyzed by Image J software. Error bars indicate the mean (SD); n = 4. P-values were determined using one-way ANOVA (*$P < 0.05$, **$P < 0.01$, ***$P < 0.001$).

Akt-mTOR pathway, belongs to oncogenic signaling transduction, could regulate multiple biological activities including transcription, protein synthesis, cell metabolism, proliferation, apoptosis, and drug resistance [26]. In this study, we found that the hyperactivity of Akt-mTOR pathway in resistant cells is accompanied with ASS1 expression. Inhibitors of Akt-mTOR pathway including MK-2206 and rapamycin, suppressed expression of ASS1 in resistant cells, which suggested ASS1 was one of the downstream elements of Akt-mTOR pathway. Growing evidences revealed the crucial role of Akt-mTOR pathway in drug resistance. PARP1 regulates EGFR-TKI resistance through inducing the PI3K/AKT/mTOR/P70S6K pathway in NSCLC [27]. The interactions between Akt-mTOR pathway and estrogen receptor enable breast cancer cell to acquire estrogen tolerance during endocrine therapy [28]. Cell proliferation and drug resistance were correlated with cell cycle-related proteins such as cyclin D1, CDK4, and CDK6. Further study demonstrated that targeting the Akt/mTOR pathway could control the protein expressions of cyclin D1 and CDK4, thus contribute to the development of drug resistance in breast cancer [29]. Besides, activation of the Akt-mTOR pathway also induces the levels of multidrug resistance-associated protein-1 (MRP1), ABCG2 and P-glycoprotein, which account for multidrug resistance in chemotherapy [30]. Currently, we have not determined these biomarkers since recombinant human arginase was modified with polyethylene glycol, which facilitates the process of cellular uptake [8].

As a targeted metabolic antitumor strategy, the outcome of arginine deprivation is related to the expression of ASS1, which is considered as the key enzyme in arginine biosynthesis and confers ferroptosis resistance in NSCLC [31]. Previous literatures have proved that ASS1 deficient cell lines are more vulnerable to arginine deprivation therapy [10,32]. For instance, Kelly with colleagues illustrated that knockdown ASS1 could keep SW1222 cells (SCLC) more susceptible to ADI-PEG20 treatment [33]. And our preliminary study also demonstrated that SCLC cells with low level of ASS1 had a metabolic vulnerability to BCT-100 treatment [10]. However, silencing of ASS1 in another SCLC cell line (H69) has little significant differences during BCT-100 exposure [10], and noncanonical pathways that regenerate arginine might account for this finding, which needs in-depth exploration. Apart from the essential role in arginine metabolism, ASS1 has multiple functions in cancer biology. For instance, ASS1 determines the efficacy of gemcitabine and docetaxel therapies in ASS1-negative tumors [34]. Mice with high ASS1 level display significantly lower objective response rates to anti-PD-1 therapy in xenograft model, and breast cancer patients with high ASS1 levels have more metastases and poor progression-free survival [35]. However, ASS1 might show different roles in hepatocellular carcinoma. High levels of ASS1 were correlated with favorable 10-year overall survival rates in a Korean cohort study, and ASS1-overexpressing HCC cell lines displayed higher cellular apoptosis, greater sensitivity to cisplatin, and diminished wound healing. Besides, endoplasmic reticulum stress contributes to ASS1 synthesis in HCC microenvironments [36].

Reactive oxygen species (ROS) homeostasis plays an important role in biological activity, and overproduction of ROS can cause oxidative damage to cellular biomolecules (e.g., lipids, proteins, DNA), thus leading to cell death. To date, substantial chemotherapeutic drugs including butaselen, ethaselen, β-Lapachone, Conoidin A, and arsenic trioxide, have been developed for several tumors [37,38]. In our study, resistant cells refused to produce excessive ROS upon BCT-100, while ROS production was significantly augmented when SESN3 was silenced, indicating SESN3 had powerful regulation on redox-catalyst system. Therefore, ROS-based tumor treatment strategies show great potential in clinical practice. Meanwhile, the crosstalk between ROS production and Akt/mTOR pathway might exist in RAW264.7 cells upon arsenic trioxide treatment [39]. Similarly, LY294002 and rapamycin were found to alleviate fibrosis by eliminating intracellular ROS and activating autophagy related biomarkers in the process of peritoneal dialysis [40]. Currently, chemokinetic therapy (CDT) is an emerging therapeutic strategy that employs biochemical reactions to produce ROS to kill tumor cells. Huang *et al* proposed a hydrogel co-loading $SO_2$ prodrug and FeGA nanoparticles for the synergistic treatment of CDT and $SO_2$ gas, which displayed promising anti-cancer effects *in vitro* and *in vivo* [41].

Nevertheless, these are several limitations in our study. The study is primarily based on in vitro experiments and animal models, lacking data from clinical patients to support the findings. Future research could incorporate clinical samples to further validate the expression of SESN3 in SCLC patients and its association with drug resistance. Although H446

is commonly used in SCLC research, the use of only one cell line may limit the generalizability of the study results, the expressions of SESN3 and ASS1/p-Akt/p-mTOR axis in another arginase-resistant SCLC line was determined in S5 Fig. Future research could be extended to other SCLC cell lines or patient samples to verify the broad applicability of the results. H446 cell line originates from a pleural effusion, which represents a metastatic sample. Based on an expression profiling tool for cancer cell lines, such as ShinyThor, H446 appears to be among the highest SESN3-expressing models [42]. It might indicate SESN3 has distinct inducible ability. For the experimental design, both total level and cleaved isoforms of caspase 3 and PARP should be tested in the future study. Besides, potential resistance mechanisms, such as drug metabolism, alterations in apoptotic pathways, or abnormalities in cell cycle regulation, need to be explored in the future.

In conclusion, our results demonstrated that high expression of SESN3 is associated with the resistance of SCLC cells to recombinant human arginase, and the underlying mechanism is involved in Akt/mTOR signal pathway following with ASS1 expression. Our study provides a scientific rational to illustrate the mechanism of drug resistance of SCLC to recombinant human arginase, and targeting Akt/mTOR/ASS1 axis might be an encouraging option in future clinical practice.

## Supporting information

**S1 Fig. The wound healing assay to determine cell migration ability of H446-BR infected with SESN3 or/and CNTN-1.** (A) Wound healing assay to assess cellular migration ability of H446-BR infected with shNEG, shSENS3, shCNTN-1, and shSESN3 plus shCNTN-1. (B) Quantitation of migration rate 48 hr in each group.
(TIF)

**S2 Fig. NAC relieves the mitochondrial membrane depolarization of H446-BR cells infected with shSESN3−1.** (A) Mitochondrial membrane depolarization shown by JC-1 staining in H446-BR cells (shNEG, NAC(5nM), shSESN3−1, shSESN3−1＋NAC) upon BCT-100 (20 mU/ml) exposure for 24 hr. (B) Quantitation of relative JC-1 monomer ratio in each group.
(TIF)

**S3 Fig. SESN3 regulates the expression of ASS1 and apoptosis.** (A)Western blot to evaluate the level of ASS1 and SESN3 in H446, H446 overexpressed with normal control, H446 overexpressed with SESN3, and H446-BR. (B) Quantitation of SESN3 and ASS1 in each group. (C) Western blot to evaluate the level of C-PARP and C-Caspase3 in presence of BCT-100 treatment (20 mU/ml, 72 hr). (D) Quantitation of C-PARP and C-Caspase3 in each group. β-Actin was used as loading control. Error bars indicate the mean (SD); n＝3. P-values were determined using one-way ANOVA (*$P < 0.05$, **$P < 0.01$, ***$P < 0.001$).
(TIF)

**S4 Fig. Akt activator SC79 recover the expressions of p-Akt, p-mTOR, and ASS1 in SESN3-knockdown H446-BR cells.** Western blot to evaluate the level SESN3, p-Akt, Akt, p-mTOR, mTOR, and ASS1 in H446-BR cells treated with shSESN3 and/or SC79(5ug/ml).β-Actin was used as loading control.
(TIF)

**S5 Fig. The expressions of SESN3 and Akt/mTOR/ASS1 axis in H526 and H526-BR cells.** Western blot to evaluate the level SESN3, p-Akt, Akt, p-mTOR, mTOR, and ASS1 in H526 and H526-BR cells.β-Actin was used as loading control. Error bars indicate the mean (SD); n＝3. P-values were determined using one-way ANOVA (*$P < 0.05$, **$P < 0.01$, ***$P < 0.001$).
(TIF)

**S1 File. WB Raw data.**
(ZIP)

## Author contributions

**Conceptualization:** Hanchao Gao, Shi Xu.

**Data curation:** Zhongqiang Zhang, Zizhe Lin, Weishan Li.

**Formal analysis:** Weishan Li.

**Methodology:** Binxiong Chen, Yueming Liu.

**Project administration:** Shi Xu.

**Supervision:** Yueming Liu, Hanchao Gao.

**Writing – original draft:** Shi Xu.

**Writing – review & editing:** Shi Xu.

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
