## [Decision Letter · Decision Letter 0]

16 Jul 2025

Dear Dr. Xu,

We look forward to receiving your revised manuscript.

Kind regards,

Alexis G. Murillo Carrasco

Academic Editor

PLOS ONE

Journal Requirements:

3. To comply with PLOS One submissions requirements, in your Methods section, please provide additional information regarding the experiments involving animals and ensure you have included details on methods of anesthesia and/or analgesia, and efforts to alleviate suffering.

5. We note that your Data Availability Statement is currently as follows: All relevant data are within the manuscript and its Supporting Information files.

6. Thank you for stating the following financial disclosure:

Guangdong Basic and Applied Basic Research Foundation(Grant No. 2024A1515012821).

7. Please update your submission to use the PLOS LaTeX template. The template and more information on our requirements for LaTeX submissions can be found at http://journals.plos.org/plosone/s/latex.

8. Please amend your list of authors on the manuscript to ensure that each author is linked to an affiliation. Authors’ affiliations should reflect the institution where the work was done (if authors moved subsequently, you can also list the new affiliation stating “current affiliation:….” as necessary).

9. We notice that your supplementary figures are uploaded with the file type 'Figure'. Please amend the file type to 'Supporting Information'. Please ensure that each Supporting Information file has a legend listed in the manuscript after the references list.

Reviewers' comments:

Reviewer's Responses to Questions

**Comments to the Author**

1. Is the manuscript technically sound, and do the data support the conclusions?

Reviewer #1: Partly

Reviewer #2: Yes

Reviewer #3: Yes

2. Has the statistical analysis been performed appropriately and rigorously?

Reviewer #1: Yes

Reviewer #2: No

Reviewer #3: Yes

3. Have the authors made all data underlying the findings in their manuscript fully available?

Reviewer #1: Yes

Reviewer #2: Yes

Reviewer #3: Yes

4. Is the manuscript presented in an intelligible fashion and written in standard English?

Reviewer #1: Yes

Reviewer #2: Yes

Reviewer #3: Yes

Reviewer #1: Authors have made an effort to understand the mechanism of resistance against the recombinant-arginase using pegylated recombinant human arginase BCT-100.

Major Comment:

It is not clear why the cell line H446 was chosen. Are these cells lack the endogenous enzymes argininosuccinate synthetase (ASS) and ornithine transcarbamylase (OTC). Or these cells are dependent on exogenous Arginine?. Thus, it is not clear if it is a cell type dependent study or can be implemented widely to arginase resistance.

Control like H446-BR cells with shNEG and shSESN3 without BCT-100 are missing throughout the experiments. So the reportd conclusions, can be misleading.

To overrule the influence of increased proliferation on increased migration observed in H446-BR if the cell cycle can be restricted before doing migration assay.

Minor Comment:

The phrase "stemness related genes", rather than stemness should be used.

Label must be given for all lanes of western blot experiments in every figure.

Tumor volume in Figure 4B should display results of all 4 conditions.

All the figures should be of high resolution. Flow cytometry data and cell percentage are barely visible.

Reviewer #2: The authors establish a BCT-100–resistant SCLC cell line (H446-BR) and show that it is cross-resistant to cisplatin/etoposide and displays enhanced migration and stem-like marker. RNA-seq points to up-regulation of Sestrin3 (SESN3); knock-down of SESN3 re-sensitizes H446-BR cells to arginase, attenuates migration/proliferation, increases ROS and mitochondrial depolarization, and restores apoptosis both in vitro and in xenografts. Mechanistically, SESN3 knock-down suppresses p-Akt, p-mTOR and ASS1, while pharmacologic Akt or mTOR inhibition likewise lowers ASS1, supporting an Akt/mTOR/ASS1 axis downstream of SESN3. Over-expression of SESN3 in parental H446 partially recreates resistance, corroborating causality. Overall, the study identifies SESN3 as a driver of arginase resistance and suggests that dual targeting of SESN3 or the Akt/mTOR/ASS1 pathway may overcome this barrier.

The strengths of this paper include (i) clear clinical rationale—arginine-auxotrophic SCLC is entering trials with BCT-100; (ii) use of complementary assays (cell viability, migration, ROS, mitochondrial potential, xenografts) to support conclusions; and (iii) mechanistic linkage to an actionable pathway (Akt/mTOR) that is readily druggable.

The weaknesses of this paper are the reliance on a single SCLC background, small xenograft cohorts, and absence of patient-derived material or publicly available clinical data to validate SESN3 up-regulation or its correlation with outcome. Several methodological descriptions (e.g., shRNA sequences, detailed statistics) and figure labelling need strengthening.

Major comments

1. Validate SESN3 expression and ASS1/p-Akt/p-mTOR levels in at least one additional SCLC line (ideally another arginase-resistant model) to show generalisability.

2. Provide human evidence: analyse publicly available SCLC datasets (e.g., CCLE, TCGA) or patient biopsies for SESN3 and ASS1 expression and correlate with reported arginase sensitivity where possible.

3. Clarify SESN3’s position in the pathway: does SESN3 interact physically or transcriptionally with Akt or ASS1? Rescue experiments using constitutively active Akt or ectopic ASS1 after SESN3 knock-down would strengthen causality.

4. Improve statistical reporting: specify exact n for each experiment, define error bars, state tests used, verify that multiple-comparison corrections were applied where appropriate.

Minor comments

1. Provide shRNA target sequences (unless proprietary restrictions are documented) to ensure reproducibility.

2. Confirm normality assumptions before applying t-tests; if data are non-parametric, use appropriate tests.

3. The term “aggressiveness” (Abstract/Results) is vague; replace with specific phenotypes (e.g., migration, proliferation).

4. Correct typographical errors (e.g., “in intro” should read “in vitro”; “Moclecular weight” in Table 1).

5. Discussion: moderate the claim that SESN3 “induces” Akt/mTOR activation unless direct evidence is provided; consider phrasing as “is associated with”.

Reviewer #3: n this current study “Sestrin3 confers resistance to recombinant human arginase in small

cell lung cancer by activating Akt/mTOR/ASS1 axis”, the authors developed a SCLC cell line

resistant to BCT-100 and determined that SESN3 contributes to the resistance phenotype.

SCLC is known for acquiring resistance to chemotherapy agents and understanding the

mechanism of resistance is vital to identify novel treatment options. The current study

requires additional work to differentiate from a previous study by this group which

determined that CNTN-1 modulates BCT-100 resistance. Please see the attached reviewer comments for major and minor revisions to the manuscript.

**Do you want your identity to be public for this peer review?** For information about this choice, including consent withdrawal, please see our Privacy Policy

Reviewer #1: **Yes: ** Sandeep Singh

Reviewer #2: **Yes: ** Wen Gu

Reviewer #3: No

---

## [Author Response · Author response to Decision Letter 1]

21 Sep 2025

All comments had been addressed in “Response to Reviewers”. Thank you very much.

---

## [Editor Report · Decision Letter 1]

29 Sep 2025

Dear Dr. Xu,

Thank you for submitting your manuscript to PLOS ONE. After careful consideration, we feel that it has merit but does not fully meet PLOS ONE’s publication criteria as it currently stands. Therefore, we invite you to submit a revised version of the manuscript that addresses the points raised during the review process.

Please include responses to all reviewers' and the editor's previous comments. It should include responses to the next paragraph. 

In particular, the main features of the small cell lung cancer (SCLC) cell line used should be clearly characterized. Based on an expression profiling tool for cancer cell lines, such as ShinyThor (https://doi.org/10.1093/bioadv/vbaf061), this cell line appears to be among the highest SESN3-expressing models. However, it originates from a pleural effusion, which represents a metastatic sample, and this should be explicitly stated in the manuscript. Additionally, since you reanalyze data from a previous study (ref19) to assess SESN3 levels, the processed dataset should be deposited in a publicly accessible repository (e.g., Gene Expression Omnibus), and the associated accession code must be included in the submission via the data availability statement. To better reflect clinical relevance, it is also recommended to include a patient-level analysis of SESN3 expression, along with other related genes, in relation to patient survival.

We look forward to receiving your revised manuscript.

Kind regards,

Alexis G. Murillo Carrasco

Academic Editor

PLOS ONE
---

## [Author Response · Author response to Decision Letter 2]

30 Sep 2025

Reply

Thanks for your very useful comment for our study. We used ShinyThor to confirm the basal level of SESN3 in H446 cell. We discuss this issue in our Discussion Part as consider it as the limitation of our study. “H446 cell line originates from a pleural effusion, which represents a metastatic sample. Based on an expression profiling tool for cancer cell lines, such as ShinyThor, H446 appears to be among the highest SESN3-expressing models [42]. It might indicate SESN3 has distinct inducible ability.”

Reply

Thanks for your comment. The raw data of RNA was deposited in Dr Jamse Ho (Professor in HKU). In this study, we used Western blot and qPCR to confirm the high level of SESN3 in H446-BR cells. However, to comply with our policy, we delete the RNA sequencing analysis statement and Ref19 accordingly. Currently, we have very limited patients with BCT-100 resistance to further confirm the relation of SESN3 to patient survival. And we have discussed this issue as regarded it as one of limitations in our manuscript.

---

## [Decision Letter · Decision Letter 2]

4 Nov 2025

Dear Dr. Xu,

Thank you for submitting your manuscript to PLOS ONE. After careful consideration, we feel that it has merit but does not fully meet PLOS ONE’s publication criteria as it currently stands. Therefore, we invite you to submit a revised version of the manuscript that addresses the points raised during the review process.

Please revise the additional reviewers' comments and evaluate the feasibility of including additional external cohorts and/or reactivating pathway assays. In cases where they cannot be possible, please discuss these potential limitations.

We look forward to receiving your revised manuscript.

Kind regards,

Alexis G. Murillo Carrasco

Academic Editor

PLOS ONE

Journal Requirements:

Reviewers' comments:

Reviewer's Responses to Questions

**Comments to the Author**

Reviewer #1: All comments have been addressed

Reviewer #2: (No Response)

Reviewer #3: All comments have been addressed

2. Is the manuscript technically sound, and do the data support the conclusions?

Reviewer #1: Yes

Reviewer #2: (No Response)

Reviewer #3: Yes

3. Has the statistical analysis been performed appropriately and rigorously?

Reviewer #1: Yes

Reviewer #2: (No Response)

Reviewer #3: Yes

4. Have the authors made all data underlying the findings in their manuscript fully available?

Reviewer #1: Yes

Reviewer #2: (No Response)

Reviewer #3: Yes

5. Is the manuscript presented in an intelligible fashion and written in standard English?

Reviewer #1: Yes

Reviewer #2: (No Response)

Reviewer #3: Yes

Reviewer #1: Authors have made sufficient corections and performed new experiments. It is adviced to please extensively proofread the manuscript and correct the phrasing of sentences.

Reviewer #2: (1) Provide a minimal but concrete public-data analysis demonstrating the prevalence and clinical context of SESN3 and ASS1 expression in SCLC—e.g., quantify transcript levels across CCLE SCLC lines and at least one patient cohort (GEO/ArrayExpress), report effect sizes and FDR-adjusted P values, and include a brief survival or subtype association if available.

(2) Establish pathway order with a mechanistic rescue: in SESN3-knockdown H446-BR (and/or H526-BR), restore signaling using constitutively active Akt or ectopic ASS1, then show recovery of mTORC1 readouts (p-S6K, p-4EBP1), ASS1 protein, and functional resistance to BCT-100.

Reviewer #3: Dear authors,

Thank you for the response to my comments. My comments on the manuscript have been addressed, and the manuscript is ready to be published. I appreciate the additional response in addressing the limitations of the study. One minor comment regards the following sentence under the section of “SESN3 silencing inhibited the cellular migration and promoted cellular apoptosis”, where the underlined words appear to be in a different font than the rest of the manuscript: “. In line with previous results, ALDH1, Oct-4, Nanog, and N-cadherin were suppressed in H446-BR cell line infected with SESN3 (Fig 3F).”

**Do you want your identity to be public for this peer review?** For information about this choice, including consent withdrawal, please see our Privacy Policy

Reviewer #1: No

Reviewer #2: **Yes: ** wen gu

Reviewer #3: No

---

## [Author Response · Author response to Decision Letter 3]

10 Nov 2025

Dear Dr. Alexis G. Murillo Carrasco,

Academic Editor

PLOS ONE,

Thank you very much for reviewing our manuscript. We received useful comments from the reviewers and would like to address them as in the following.

Manuscript title: Sestrin3 confers resistance to recombinant human arginase in small cell lung cancer by activating Akt/mTOR/ASS1 axis.

Reviewer(s)' Comments to Author:

Reviewer #1: Authors have made sufficient corections and performed new experiments. It is adviced to please extensively proofread the manuscript and correct the phrasing of sentences.

Reply: Thanks for your kindly review.

Reviewer #2: (1) Provide a minimal but concrete public-data analysis demonstrating the prevalence and clinical context of SESN3 and ASS1 expression in SCLC—e.g., quantify transcript levels across CCLE SCLC lines and at least one patient cohort (GEO/ArrayExpress), report effect sizes and FDR-adjusted P values, and include a brief survival or subtype association if available.

Reply: Thanks for your valuable advice. We made gene correlation analysis between SESN3 and ASS1 by GePIA cancer database (Figure 1). It seems that these two genes have not significant relevance within normal lung cancer. Besides, the SESN3 and ASS1 expressions in SCLC cell lines vary significantly (Figure 2). However, these data were analyzed in normal SCLC, which may differ from BCT-100 resistant context. Due to the limited clinical samples, currently we cannot explore the relation of genes level and brief survival, which is one of limitations in this study.

(2) Establish pathway order with a mechanistic rescue: in SESN3-knockdown H446-BR (and/or H526-BR), restore signaling using constitutively active Akt or ectopic ASS1, then show recovery of mTORC1 readouts (p-S6K, p-4EBP1), ASS1 protein, and functional resistance to BCT-100.

Reply: Thanks for your suggestion. In our current study, we already found SESN3-knockdown could reserve phenotype of drug resistance in H446-BR cells. To confirm our findings, we used Akt activator SC79 in H446-BR cells transfected with shSESN3, with recovery of p-Akt, p-mTOR, and ASS1 (S4 Fig).

Reviewer #3: Dear authors,

Thank you for the response to my comments. My comments on the manuscript have been addressed, and the manuscript is ready to be published. I appreciate the additional response in addressing the limitations of the study. One minor comment regards the following sentence under the section of “SESN3 silencing inhibited the cellular migration and promoted cellular apoptosis”, where the underlined words appear to be in a different font than the rest of the manuscript: “. In line with previous results, ALDH1, Oct-4, Nanog, and N-cadherin were suppressed in H446-BR cell line infected with SESN3 (Fig 3F).”

Reply: Thanks for your kindly review. We revised the font accordingly.

Sincerely,

Shi Xu

Please address the correspondence to:

Shi Xu, PhD

Associate researcher

Shenzhen Longhua District Central Hospital,

Shenzhen,

Guangdong,

China

Email: xushi_cn@163.com Tel: (86) 755 2801 4167

Figures to address reviewer’s comments.

Figure 1 Gene correlation analysis between SESN3 and ASS1.

Figure 2 Quantify transcript levels of ASS1 and SESN3 across CCLE SCLC lines via GEO database (Accession: GSE193455).

---

## [Decision Letter · Decision Letter 3]

28 Nov 2025

Sestrin3 confers resistance to recombinant human arginase in small cell lung cancer by activating Akt/mTOR/ASS1 axis

PONE-D-25-29381R3

Dear Dr. Xu,

We’re pleased to inform you that your manuscript has been judged scientifically suitable for publication and will be formally accepted for publication once it meets all outstanding technical requirements.

Kind regards,

Alexis G. Murillo Carrasco

Academic Editor

PLOS ONE

Additional Editor Comments (optional):

Reviewers' comments:

Reviewer's Responses to Questions

**Comments to the Author**

Reviewer #1: All comments have been addressed

Reviewer #2: All comments have been addressed

Reviewer #3: All comments have been addressed

2. Is the manuscript technically sound, and do the data support the conclusions?

Reviewer #1: Yes

Reviewer #2: Yes

Reviewer #3: Yes

3. Has the statistical analysis been performed appropriately and rigorously?

Reviewer #1: Yes

Reviewer #2: Yes

Reviewer #3: Yes

4. Have the authors made all data underlying the findings in their manuscript fully available?

Reviewer #1: Yes

Reviewer #2: Yes

Reviewer #3: Yes

5. Is the manuscript presented in an intelligible fashion and written in standard English?

Reviewer #1: Yes

Reviewer #2: Yes

Reviewer #3: Yes

Reviewer #1: All comments are addressed. I do not have any further observations. Thanks.

Reviewer #2: The authors have addressed the core requests in spirit: they added a minimal public-data look (GePIA correlation and CCLE/GEO expression ranges) and performed a pharmacologic rescue (SC79) in the SESN3-knockdown background that restores p-Akt/p-mTOR/ASS1. They also include a second model (H526/H526-BR) in the supplement, which helps generalizability. These additions substantively strengthen the story.

Reviewer #3: (No Response)

**Do you want your identity to be public for this peer review?** For information about this choice, including consent withdrawal, please see our Privacy Policy

Reviewer #1: No

Reviewer #2: No

Reviewer #3: No

---

## [Editor Report · Acceptance letter]

PONE-D-25-29381R3

PLOS One

Dear Dr. Xu,

I'm pleased to inform you that your manuscript has been deemed suitable for publication in PLOS One. Congratulations! Your manuscript is now being handed over to our production team.

Kind regards,

on behalf of

Dr. Alexis G. Murillo Carrasco

Academic Editor

PLOS One